# Adipokines Level and Cognitive Function—Disturbance in Homeostasis in Older People with Poorly Managed Hypertension: A Pilot Study

**DOI:** 10.3390/ijerph19116467

**Published:** 2022-05-26

**Authors:** Agnieszka Kujawska, Sławomir Kujawski, Mariusz Kozakiewicz, Weronika Hajec, Małgorzata Kwiatkowska, Natalia Skierkowska, Jakub Husejko, Julia L. Newton, Paweł Zalewski, Kornelia Kędziora-Kornatowska

**Affiliations:** 1Department of Human Physiology, Collegium Medicum in Bydgoszcz, Nicolaus Copernicus University in Toruń, 85-092 Bydgoszcz, Poland; 2Department of Exercise Physiology and Functional Anatomy, Ludwik Rydygier Collegium Medicum in Bydgoszcz Nicolaus Copernicus University in Torun, Świętojańska 20, 85-077 Bydgoszcz, Poland; skujawski@cm.umk.pl (S.K.); p.zalewski@cm.umk.pl (P.Z.); 3Department of Geriatrics, Collegium Medicum in Bydgoszcz, Nicolaus Copernicus University in Toruń, 85-094 Bydgoszcz, Poland; markoz@cm.umk.pl (M.K.); weronika.topka.bydg@gmail.com (W.H.); malgorzata.gajos0904@gmail.com (M.K.); nataliaskierkowska1@gmail.com (N.S.); kubahusejko@gmail.com (J.H.); kasiakor@interia.pl (K.K.-K.); 4Population Health Sciences Institute, The Medical School, Newcastle University, Newcastle-Upon-Tyne NE2 4AX, UK; julia.newton@ncl.ac.uk; 5Department of Experimental and Clinical Physiology, Laboratory of Centre for Preclinical Research, Warsaw Medical University, 1b Banacha Street, 02-097 Warsaw, Poland

**Keywords:** trophic factor, hormones, gerontology

## Abstract

Aim: To explore the network relationship between cognitive function, depressive symptom intensity, body composition, proxies of cognitive reserve, trophic factor, adipokines and myokines, physical performance and blood pressure in a group of older people with poorly managed hypertension (PMHTN) compared to a normotensive (NTN) group. Materials and methods: History of hypertension and blood pressure level were examined in older participants. Thirty-one subjects diagnosed with PMHTN (history of hypertension diagnosis and values of sBP or dBP over 140/90 mmHg) and eighteen NTN (lack of history of hypertension and sBP and dBP lower than 140/90 mmHg) participated. Participants completed physical and cognitive function assessments: including the Mini–Mental State Examination (MMSE), Montreal Cognitive Assessment (MoCA) and its two subtests Delayed Recall (DR) and Verbal Fluency (VF) and Trail Making Test Part B (TMT B). Factors associated with cognitive functioning: age, years of education, cognitive and travel activity were assessed using a questionnaire. Visceral fat was determined by bioimpedance testing and gait velocity and agility assessed using an Up and Go test. To summarize the strength and direction (negative or positive) of a relationship between two variables, Spearman’s rank correlation coefficient was used. Then, network graphs were created to illustrate the relationship between variables. Node strength (number of edges per node), neighbourhood connectivity (the average connectivity of all the neighbours of a node), stress (the number of shortest paths passing through each node) were compared in network from PMHTN group to network from NTN group. Results: Neighbourhood connectivity and stress were significantly higher in of the PMHTN network compared to NTN (6.03 ± 1.5 vs. 4.23 ± 2.5, *p* = 0.005 and 118.21 ± 137.6 vs. 56.87 ± 101.5, *p* = 0.02, accordingly). Conclusion: In older subjects with poorly managed hypertension, dyshomeostasis was observed, compared to normotensive subjects.

## 1. Introduction

Untreated hypertension might be related to dyshomeostasis observed in many physiologic systems [1]. Hypertension management in older people has shown correlation between higher blood pressure (BP) with higher prevalence of many disorders, including stroke, coronary heart disease, heart failure, peripheral artery disease, end-stage renal disease and sudden death [2]. In addition, it was observed that hypertension has a negative influence on cognitive function in older people [2]. A range of health complications might occur with poor adherence to hypertension treatment, including increased vascular stiffness, left ventricular hypertrophy, cognitive function decline, decreased quality of life and work productivity [3].

Hypertension is characterized by pleiotropic negative health consequences, but also has many potential risk factors. The latter group might include: low levels of physical activity level, diet with low quantity of vegetables and fruits, body composition (obesity and overweight) and low educational level [4,5]. These variables that are recognized as individual as risk factors and consequences of hypertension may also be related to each other in a mutual way. In addition, some risk factors might indirectly influence other risk factors, what in turn would create a cluster of inter-related risk factors interfering with each other and hypertension development.

Body composition changes in older people with a trend towards a decrease in lean body mass [6]. In turn, low muscle mass and sarcopenic obesity is related to worse functional capacity [7]. Obesity is associated with many comorbidities, particularly, hypertension, decreased physical function and worse cognitive function [8,9]. Recent studies have shown that both adipose and muscle tissues are hormonally active. The level of adipokines has been shown to be related to obesity and body fat distribution [10]. Moreover, adipokine profile is more directly related to pathological mechanisms of metabolic syndrome than the level of adipose tissue [10].

Myokines also might be related to obesity-related pathology. Irisin belongs to adipomyokines group, because it is both adipokine and myokine [11]. Irisin might play a role in the pathophysiology of hypertension [12]. In addition, adipokines and myokines might be related to cognitive function in older people. Higher levels of adiponectin were noted in mild to moderate Alzheimer’s dementia compared to patients in the prodromal phase of the disease [13].

As noted by other Authors, high BP in older people is a complex condition [2]. Aging negatively influence on body composition by a decrease in muscle mass and increase in visceral fat. As a consequence, those changes might lead to disturbances in the pattern of hormone production by those tissues, which in turn might negatively impact both blood pressure homeostasis and cognitive function [6,8,9]. Similarly, low physical activity level might be a risk factor for both cognitive function decline and hypertension [4,5,14]. In addition, the level of education might be related to hypertension development, but is also one indicator of cognitive reserve [15]. Factors related to cognitive reserve could also include levels of physical and cognitive activity [10]. On the other hand, a higher severity of depression symptoms may lead to decreased physical activity levels and cognitive function in older people [16]. In contrast, high levels of physical activity could serve a protective role against the emergence of depression in older people [17].

Authors suggest that new therapeutic strategies for chronic disorders could be implemented based on the analysis of the architecture of aberrant networks [18]. Moreover, disease classification used currently, might have a tendency to omit the interconnected character of many disorders [19]. In response, a systems-based network framework for defining human disease has been proposed [19].

Therefore, we in this explorative study we have compared networks of cognitive function, depression symptoms intensity, body composition, proxies of cognitive reserve, trophic factor, adipokines and myokines, physical performance and blood pressure in group of older people with poorly managed hypertension (PMHTN) comparing to normotensive (NTN) group. We hypothesize that these variables are interconnected to each other and that difference in the pattern of this interconnection might reveal dyshomeostasis in PMHTN group.

## 2. Materials and Methods

### 2.1. Study Group

Participants were recruited into this pilot study based on a regional TV and radio advertising campaign, during health-promoting lectures at Collegium Medicum University and many older people’s organizations in Bydgoszcz, in Day Care Centers for the Elderly, and at various meeting-groups for older people. Study marketing material included literature and information describing an opportunity to participate free-of-charge in a study measuring physical, physiotherapeutic, dietary, social and cognitive measures for people 60 years old and over. The recruitment period of participants into the current study took place between November 2015 and February 2018. Age under 60 years old was the only excluding factor from participation in study. Examination was conducted in the Department of Geriatrics, Collegium Medicum University Hospital in Bydgoszcz, Poland. The study was approved by the Ethics Committee, Ludwik Rydygier Memorial Collegium Medicum in Bydgoszcz, Nicolaus Copernicus University, Torun (KB 340/2015). Written, informed consent was obtained from all participants.

### 2.2. Assessment Methods

#### Cognitive Tests

Ninety-seven percent of cognitive function tests were conducted by the same person (SK). General cognitive function level was assessed using Mini–Mental State Examination (MMSE) and Montreal Cognitive Assessment (MoCA), while executive functioning domain: visuospatial skills, task switching and working memory was examined using Trail Making Test Part B (TMT B) [20].

Both MMSE and MoCA are cognitive function screening 30-point questionnaires. A high scores indicate better cognitive performance [21,22]. In the case of MoCA. Results of two subtests (Verbal Fluency (VF) subtest and Delayed Recall (DR) were considered during the analysis, besides of the total score.

In the DR subtests subjects were exposed to the list of 5 words two times in a row. Subjects were asked to memorize words, and then the recall was performed after a few minutes. Result in the DR subtest that might range between 0 to 5, indicates the total number of words that subjects were able to recall after free recall and category and list cues.

Score in the VF subtest indicates the number of words in Polish starting with letter “S” which are not own nouns (conjugation was prohibited) that subject was able to express in one minute Therefore, higher score indicate better verbal fluency [23]. Trail Making Test part B score is a time spend on test completion. Therefore, the higher the score, the worse executive function [20].

### 2.3. Functional Performance Assessment

The Timed up-and-go [24] test serve as an indicator of gait speed and balance in dynamic manner. Subjects are asked to complete the task as fast as possible. The task is to get up from chair and go straight ahead to and around a marker placed on floor three meters away from the chain and get back and sit on the chair again.

To measure aerobic capacity of patients, six-minute walk test (6MWT) was performed [25]. To reduce time spent turning the test was performed on flat corridor with distance of 50 m. Participants were asked to maintain the same velocity during the whole test while walking as fast as possible. To exclude competition between subjects, most performed the test alone, otherwise sufficient interval between consecutive participants was maintained [26].

In addition, four subtests from Fullerton Functional Fitness Test were used: arm curl test and 30-s chair stand, sit and reach test [27].

To measure upper limb strength arm curl test was performed using two types of weight: 2 kg for women and 3.5 kg for men. During the test, subject was holding the weight in comfortable grip, while sitting position on the chair with a backrest. We avoided full extension to the side of the chair, due to the risk of the injury experience in subjects with under-diagnosed osteoporosis. Therefore, we decided to start the movement from weight positioned on thigh. Then, supinating during flexion was advised so that the palm of the hand faced the biceps brachii muscle at the end of concentric phase, if the initial position of palm was in directed in another way. Left- and right-hand strength was assessed separately [27]. Mean results based on number of repetitions performed by participants using left and right hand were analyzed only.

To measure lower limbs strength 30-s chair stand test was performed on the chair with a backrest. Test program contains standing from sitting position to a standing with full extension in knees and hips, without pushing off with the arms. Tests score was the number of repetitions consisted of standing and sitting phase performed in 30 s [27].

### 2.4. Body Composition Analysis

Tanita BC-545 body-fat analyzer served to examine body mass and body composition. Participants were weighed in light clothing. Weighing accuracy is 0.1%. Body composition indicators (Body Water Content (BWC), body fat (%), visceral fat (units), muscle mass (kg) were measured using bioelectric impedance analysis (BIA). Based on results basal metabolic rate (BMR) measured in kcals was calculated using a built-in algorithm. Participants were asked to stand on electrodes on sole of foot and holding electrodes in hands. Respondents themselves gave information about height to reduce amount of time spend on examination and body mass index (BMI) was calculated in accordance to World Health Organization (WHO) recommendations [28].

### 2.5. Activity Level Questionnaire and Physical Examination

Frequency of current physical, mental and social activities was assessed using questionnaire, described in more detail previously [21]. Frequency was coded into seven categories (“never”, “once a year”, “several times a year”, “1–2 times a month”, “once a week”, “few times a week”, and “daily”). Then, variables based on total score of overall activity was calculated and analyzed.

### 2.6. Blood Pressure Examination

Blood pressure assessment was done in the doctor’s office. Applied methodology was based on the 2013 ESH/ESC Practice Guidelines [29]. Systolic Blood Pressure (sBP) and Diastolic Blood Pressure (dBP) were measured as one followed by another upper limb. Every examination was taken after at least 5 min of sitting in upright position. Mean value for each sBP and dBP from these two measurements were analyzed. Pulse pressure (PP) was calculated using formula:PP = sBP − dBP

Hypertension was examined during patients history examination. Participants were asked to recall the history of hypertension diagnosis and its treatment. Before examination, subjects were reminded during a phone call to take a list of medications as well as documents regarding patients history, to facilitate the process of data collection.

Subjects with poorly managed hypertension were distinguished based on history of hypertension diagnosis and values of sBP or dBP over 140/90 mmHg during assessment in the above study. The NTN group was distinguished based on a lack of history of hypertension and sBP and dBP lower than 140/90 mmHg, respectively.

### 2.7. Emotional State Assessment

Trained technician assessed the severity of depression using the 15-item Geriatric Depression Scale (GDS) [30]. The shorter version, composed of 15 questions binary (yes/no) responses could be applied both in older people with intact and decline of cognitive function [31]. The total score might range from 0 to 15 points. The higher the score, the higher the severity of depression.

### 2.8. Biochemical Examination

Samples of venous blood were taken from the cubital vein to 6 mL tubes with a coagulation activator. Blood for serum was left for 20 min to clot and then centrifuged (3000× *g* for 15 min) and separated into portions (300 μL) for sterilized tubes, where they were stored at −80 °C until designation of adipokines. ELISA enzyme immunoassays for levels determination of Adiponectin [ug/mL], Irisin [ug/mL], Vaspin [ng/mL], Vsisfatin [ng/mL], Insulin-like growth factor 1 (IGF-1) [ng/mL], Insulin-like growth factor-binding protein 3 (IGFBP-3) [ng/mL] were examined using validated kits from BioVender, Brno, Czech Republic.

### 2.9. Statistical Analysis

Descriptive statistics were presented as mean and standard deviation (SD). To test the assumption of normality Shapiro–Wilk test was used and histograms were visually inspected. Levene’s test was used to assess the equality of variances. If the assumptions on normality of distribution and equality of variances were met, then independent T-tests was used, otherwise Mann–Whitney U was applied to examine between-group differences. The α was set at 0.05 and thus a *p*-value lower than 0.05 was considered as statistically significant. Effect size (r) was calculated for significant comparisons [32].

To summarize the strength and direction (negative or positive) of a relationship between two variables, Spearman’s rank correlation coefficient was used. Cytoscape software version 3.8.1 (The Cytoscape Consortium, 4221 Hill St, San Diego, CA 92107, USA) was used to perform the network analysis [33]. Nodes are illustrated as dots. The variables were grouped according to categories illustrated by colour of nodes: cognitive function (red node), depression symptoms intensity (yellow), body composition (cyan), trophic factor, adipokines and myokines (pink), physical performance (green), blood pressure (pink) and blue for social variables: years of education, cognitive and physical activity level. The size of the dots next to the variable names indicates the quantity of significant correlation coefficients with other variables, i.e., the higher the quantity, the bigger the dot. Direction of correlation is indicated by colour of edges (connections between nodes indicated by dots): blue illustrates negative while red positive correlation. Effect size of correlation is illustrated by edge (connection) width and colour intensity. Strength of correlation without negative sign served as an input to Prefuse Force Directed Layout algorithm implemented in Cytoscape software. Node strength (number of edges per node), neighbourhood connectivity (the average connectivity of all the neighbours of a node), stress (the number of shortest paths passing through each node) were compared using Mann–Whitney U test. We have selected those variables based on previous study on dyshomeostasis [34]. In addition, qualitative analysis of between-network differences was conducted.

## 3. Results

The group with poorly managed hypertension consisted of 31 participants (2 males), normotensive group consisted of 18 participants (1 male).

As Table A1 shows, the PMHTN group was significantly older: 70.84 ± 6.2 vs. 66.17 ± 4.8 years old in the NTN group, Z = 2.63, *p* = 0.01, r = 0.38. Subjects with PMHTN had higher BMI (27.68 ± 3.9 vs. 24.02 ± 3.9, *p* = 0.01, r = 0.4), and body fat percentage (35.76 ± 6.2 vs. 28.64 ± 9, *p* = 0.004, r = 0.44). Moreover, the visceral fat level was higher in the PMHTN group 10.63 ± 2.6 vs. 7.41 ± 2, *p* = 0.0002, r = 0.59. Trivial differences in blood pressure were also observed.

Figure 1a,b, Figure 2 and Figure 3 represent network analysis in both groups, and separately in PMHTN and NTN groups, respectively. All nodes are shown in both networks, to reveal nodes with and without edges, therefore, number of nodes is the same in both groups (*n* = 30).

Quantitative analysis of between-network differences showed no statistically significant difference (*p* > 0.05) in centrality of network measures (betweenness and closeness). Neighbourhood connectivity (the average connectivity of all the neighbours of a node), was significantly higher in network of hypertensive group (5.86 ± 1.8 vs. 4.2 ± 2.5 in NTN group, Z = −2.81, *p* = 0.005, r = −0.36). Moreover, stress (the number of shortest paths passing through each node) was significantly higher in network of hypertensive group (114.27 ± 136.9 vs. 56.87 ± 101.5 in NTN group, Z = −2.14, *p* = 0.03, r = −0.28). Number of edges per node was significantly higher in network in hypertensive group (4.67 ± 2.6 vs. 3.33 ± 2.6 in NTN group, Z = −2.02, *p* = 0.04, r = −0.26). However, no significant differences in edge betweenness (the number of shortest paths between vertices that contain the edge) were observed (*p* > 0.05).

In qualitative analysis of between-network differences one can observe that chronological age was connected to more nodes in hypertensive group than in NTN one. Depression symptoms intensity was negatively related with MMSE score in NTN group, while no such connection was observed in the hypertensive group. In contrary to hypertensive group, in the NTN group there was no relationship between pulse pressure with vaspin. Years of education and level of physical activity, vaspin and irisin were not related to any other factors in the NTN group. Overall, trophic factor, adipokines and myokines have more connections to other factors in the hypertensive group.

## 4. Discussion

In the current pilot study we have noted differences in both general topology and amongst specific nodes and their interactions within networks. This illustrates interconnections between selected hormones, social factors, physical and cognitive performance and blood pressure in older subjects with poorly managed hypertension compared to normotensive. Liu et al. described the network analysis of comorbidities in hypertension patients [35] where coronary heart disease, hyperlipemia, atherosclerosis, and diabetes mellitus belonged to the core of the network. To our best knowledge, the current study is the first to examine differences in the networks of adipokines level and cognitive function in older subjects with normal blood pressure vs. poorly managed hypertension.

Observing networks may allow exploration of the mechanisms that underly certain human disorders and these might be subcategorized into several dimensions, such as metabolic, disease and social networks.

It is likely these three dimensions are highly interconnected which would be obvious when focusing on relationships between a subset of variables from one dimension in a given analysis. In addition, even when exploring factors from one of those dimensions only, the complex pattern of interconnections between analyzed variables might not be evident, if a network approach is not applied. For instance, in the case of research solely on risk factors for hypertension, the standard way of analysis of a relationship between a particular risk factor and hypertension might omit the very important aspect of relationship between risk factors itself [35]. In the current study, we have observed differences in the overall topology of the network and interconnection between adipokines, myokines, body composition, cognitive function, proxies of cognitive reserve, physical performance and blood pressure in older patients with poorly managed hypertension vs. normotensive, what might indicate the presence of homeostasis disturbance in the former group.

### 4.1. Relationship between Trophic Factor, Adipokines and Cognitive Function in Older People

We have observed a positive relationship between adiponectin and the verbal delayed recall subtest score. A recent review suggested a causal role of adiponectin in neuroplasticity promotion in the hippocampus [36]. Based on the current results, PMHTN is associated with a disruption of this connection. Moreover, higher level of vaspin were related to a worse performance in MMSE and TMT B in the PMHTN group. Higher levels of vaspin were observed in older people with frailty compared to a control group without frailty [37]. Furthermore, frailty is associated with poor global cognitive function in older people [38].

IGF-1 and IGFBP-3 were positively related to level of cognitive activity in the NTN group, while no such connection was observed in the PMHTN group in the current study. Zhu et al. observed a 41% reduction of cognitive impairment risk in a group with high level of participation in leisure activities [39]. Al-Delaimy et al. noticed that IGF-1 was positively related to MMSE and Verbal Fluency, and IGFBP-1 was negatively correlated with MMSE in older men [40].

### 4.2. Relationship between Adipokines and Blood Pressure in Older People

In the current study, positive correlations between visfatin and mBP and sBP were observed in the NTN group, while negative correlation between vaspin and PP was observed in PMHTN group. Yamaleyeva et al. have shown that apelin plays a role in blood pressure regulation [41]. Therefore, further studies should explore the potential relationship between adipokines and hypertension pathology.

### 4.3. Relationship between Trophic Factor, Adipokines, Body Composition and Functional Performance

In the NTN group, a positive relationship between adiponectin level with BMR was noted in the current study. BMR is correlated positively with muscle mass. Therefore, Baker et al. concludes that a higher serum adiponectin level is associated with lower skeletal muscle mass are contrary to our findings [42]. In the hypertensive group the relationship between body composition and functional performance with biochemical factors was more complex, comparing to NTN group. Zhao et al. suggested irisin level as a marker of sarcopenia [43]. In contrast, we have observed a negative relationship between irisin and lower and upper limb strength. A positive relationship between vaspin and BMI and body fat were observed in the PMHTN group. This is in keeping with results from the study of Pazgan-Simon et al., where vaspin levels were higher in patients with BMI > 30 kg/m^2^ [44]. A negative relationship between vaspin and lower limbs strength was observed, in line with previous studies, in which low vaspin levels were related to a high fitness level [45]. A negative relationship between visfatin and the Up and go test was observed in the PMHTN group. Moreover, a negative relationship between IGFBP-3 and vaspin, as well as between IGF-1 and vaspin, were noted. Levels of vaspin [45] and visfatin in males [46] are positively related to fat in body composition analysis. Aging is related to an increase in body total fat mass and an accompanying decrease in lean mass and bone density, that are independent from general and physiological fluctuations in BMI [47]. Therefore, the relationship between adipokines with functional performance might be a spurious relationship, where, in fact, adipokines would be positively related to fat mass, and fat mass negatively related to muscle mass, which eventually would be directly related to functional performance. However, the relationships between IGF-1 and IGFBP-3 with adipokines and functional performance is worth exploration in further studies.

### 4.4. Implications for Clinical Practice

It seems that the observation that there is a simple unilateral relationship between a set of risk factors with the increased risk of hypertension might not reveal the whole mechanism underlying hypertension development in older people. Previous study showed the importance of taking age into account in models predicting cardiovascular risk related to hypertension, as young women are characterized by low risk even in the case of presence of more than one major risk factor [48]. Aging itself could be related to increased risk of multiple chronic disorders and risk factors, as chronic inflammation, frailty and cardiovascular disease [49]. It is important to take into account the multiple other relationships between the risk factors with themselves and multiple disorders, outwith hypertension only. In fact, it seems that there is a very complex relationship between education level, cognitive function, BMI, body composition, and pattern of adipokines and myokines expression. For instance, low education level could serve as a risk factor in older subjects for both cognitive decline [50], lower nutrition awareness [51], lower frequency of undertaking intensive physical activity [52].

Therefore, the network relationships between risk factors and the negative health consequences of hypertension and other disorders should be taken into account in the further research. In terms of the approach taken in communication with patients via public health platforms, information on the pleiotropic health benefits of some non-pharmacological therapies addressing factors from the network variables explored in this study should be underlined. Interventions focused on changes in the physical activity level [53] and diet with caloric deficit [54] might lead to reverse of age-related negative changes in body composition. In addition, a physical exercise program might improve cognitive function in older people [55], and lead to a reduction in depression severity [17]. Furthermore, programs composed of both, aerobic training, resistance training and its combination might reduce blood pressure of −6 mmHg in dBP and 2–3.5 mmHg in dBP in older subjects [56]. Potentially, some of those benefits induced by non-pharmacological therapies might be partially mediated by alternations in myokines and adipokines level [57,58].

### 4.5. Study Limitations

One of the potential limitations of this study, is the relatively small sample size of participants, which arguably compromises the inferences and repercussions of the work in clinical practice. Furthermore, it is important to acknowledge that the two compared groups are not well balanced in terms of sample sizes, with eighteen participants (one male) in NTN and thirty-one subjects (two males) with PMHTN. Participants in the NTN group were significantly younger, which also might have influenced the results. However the effect size of this difference was small. In future studies the sample size of both subgroups should be increased as well as better balanced in terms of characteristics of the main demographic variables as age, sex and weight. Additional limitations to recognize include; reported height by participants was included rather than measured height which might led to bias in BMI calculation. In addition, the physical and cognitive activity level was measured using questionnaires, with reference to the frequency of particular activities only, without providing information on duration or intensity. In addition, there are many potential variables that might affect blood pressure, the present study did not represent an exhaustive list and additional variables that might have an affect should not be discounted. Further studies should incorporate examination of leptin as well as low-density lipoprotein (LDL), high-density lipoproteins (HDL), triglycerides. Due to the explorative nature of the above study, sample calculation was not made. However, to our knowledge, this is the first study to examine difference in NTN compared to PMHTN older people using network approach.

## 5. Conclusions

In the current pilot study, we have noted a disturbance of homeostasis in older subjects with poorly managed hypertension, compared to normotensive subjects. Our results of an exploratory analysis show differences in the network structure underlying the relationship between trophic factor, adipokines, myokines, body composition, proxies of cognitive reserve, physical performance and cognitive function in a group of subjects with poorly managed hypertension and normotensive in both general topology, amongst specific nodes and their interactions.

## Figures and Tables

**Figure 1 ijerph-19-06467-f001:**
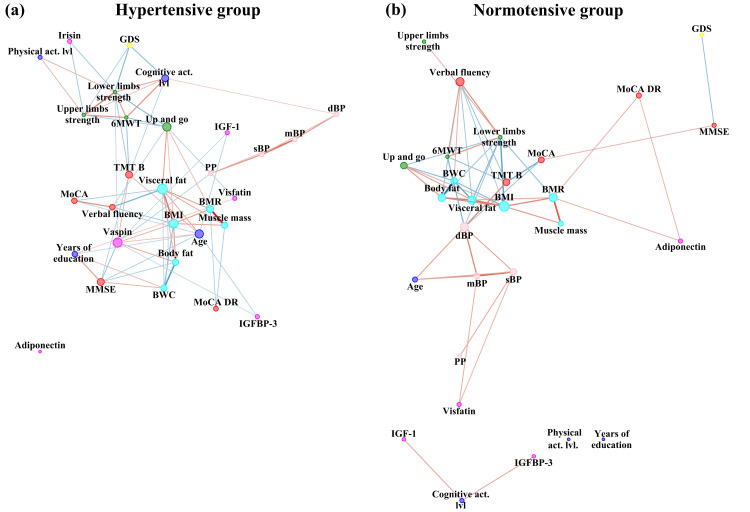
Network analysis comparison in PMHTN and NTN groups. PMHTN group network is presented on left side (**a**), NTN group on the right side (**b**). sBP—systolic blood pressure, dBP—diastolic blood pressure, mBP—mean blood pressure, PP—pulse pressure, MMSE—Mini–Mental State Examination, MoCA—Montreal Cognitive Assessment, MoCA DR—Montreal Cognitive Assessment Delayed Recall, TMT B—Trail Making Test part B, BMR—Basal Metabolic Rate, BWC—Body Water Content, BMI—Body Mass Index, 6MWT—six-minute walk test, IGF-1—Insulin-like growth factor 1, GDS—Geriatric Depression Scale.

**Figure 2 ijerph-19-06467-f002:**
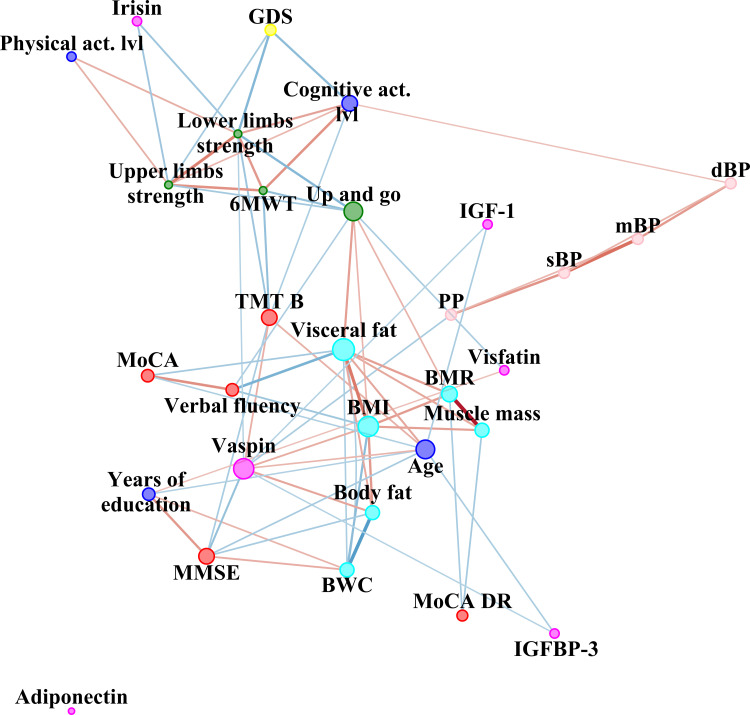
Network analysis in PMHTN group. sBP—systolic blood pressure, dBP—diastolic blood pressure, mBP—mean blood pressure, PP—pulse pressure, MMSE—Mini–Mental State Examination, MoCA—Montreal Cognitive Assessment, MoCA DR—Montreal Cognitive Assessment Delayed Recall, TMT B—Trail Making Test part B, BMR—Basal Metabolic Rate, BWC—Body Water Content, BMI—Body Mass Index, 6MWT—six-minute walk test, IGF-1—Insulin-like growth factor 1, GDS—Geriatric Depression Scale.

**Figure 3 ijerph-19-06467-f003:**
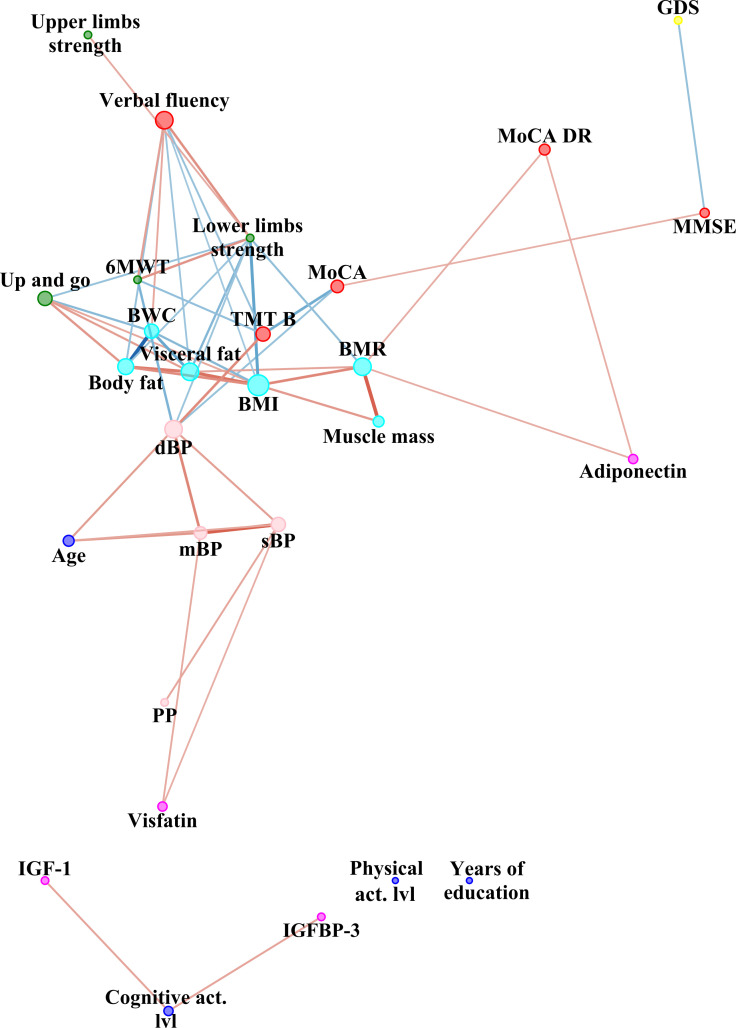
Network analysis in NTN group. sBP—systolic blood pressure, dBP—diastolic blood pressure, mBP—mean blood pressure, PP—pulse pressure, MMSE—Mini–Mental State Examination, MoCA—Montreal Cognitive Assessment, MoCA DR—Montreal Cognitive Assessment Delayed Recall, TMT B—Trail Making Test part B, BMR—Basal Metabolic Rate, BWC—Body Water Content, BMI—Body Mass Index, 6MWT—six-minute walk test, IGF-1—Insulin-like growth factor 1, GDS—Geriatric Depression Scale.

## Data Availability

Individual data are available from the corresponding author S.K. on request.

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
