# Peer review of "Adipokines Level and Cognitive Function—Disturbance in Homeostasis in Older People with Poorly Managed Hypertension: A Pilot Study"

_ijerph, 2022, doi:10.3390/ijerph19116467_

Round 1

Reviewer 1 Report

The authors complied with the corrections. I also emphasize that if there is a possibility to put in the title or in the methodology - the statement that it is a pilot study / or an initial study. I hope that the authors continue to study this topic with a greater number of participants.

Author Response

The authors complied with the corrections. I also emphasize that if there is a possibility to put in the title or in the methodology - the statement that it is a pilot study / or an initial study. I hope that the authors continue to study this topic with a greater number of participants.

Re 1: We are very grateful to the Reviewer for their comments. We have added information that it is a pilot study  to the title as suggested.

We agree, that it is important  to pursue further research in this topic and we are grateful to the reviewer for their support.

Reviewer 2 Report

The authors used network analysis to compare 31 elderlies with uncontrolled hypertension and 18 normotensive controls in terms of cognitive function, depressive symptoms, body composition, trophic factors, adipokines, myokines, physical performance and blood pressure, in order to establish whether hypertensive patients show a clustering and interaction of risk-factors compared to normotensive. 

Major points:

-The abstract should be substantially revised as it is not very clear and has unblanced sections (most of it are methods). I would suggest to include mentions of the methods used to create and compare networks in the two groups (eg. neighbourhood connectivity), this is an uncommon and not immediate method of analysis and it must be addressed in the abstract. Similarly, the results section is very brief and contains a series of numerical data that are difficult to interpret without an adequate explanation of neighbourhood connectivity and stress.

-The English written language needs to be substantially improved.

-The manuscript needs to be checked for typos and similars (missing or inappropriate use of parenthesis), which are too frequent.

-Among adipokines, do the authors have data on leptin? This would be useful as it is a strong determinant of BP and also of glucose homeostasis.

-The recruitment period is not stated.

-It is not clear why and how the variables included in the network analysis were chosen.

-The discussion should begin with a brief summary of the main findings, not with the results of another paper.

Unfortunately, although of potential clinical interest, the key message of the paper is not clear and the authors focus too much on the possible significance of each risk factors without offering a comprehensive interpretation of the results.

Author Response

The authors used network analysis to compare 31 elderlies with uncontrolled hypertension and 18 normotensive controls in terms of cognitive function, depressive symptoms, body composition, trophic factors, adipokines, myokines, physical performance and blood pressure, in order to establish whether hypertensive patients show a clustering and interaction of risk-factors compared to normotensive. 

Major points:

-The abstract should be substantially revised as it is not very clear and has unblanced sections (most of it are methods). I would suggest to include mentions of the methods used to create and compare networks in the two groups (eg. neighbourhood connectivity), this is an uncommon and not immediate method of analysis and it must be addressed in the abstract. Similarly, the results section is very brief and contains a series of numerical data that are difficult to interpret without an adequate explanation of neighbourhood connectivity and stress.

Re 1: We are grateful for the Reviewer’s comment. The abstract has been reviewed and we hope that the reviewer agrees that it has now been improved.

-The English written language needs to be substantially improved.

-The manuscript needs to be checked for typos and similars (missing or inappropriate use of parenthesis), which are too frequent.

Re 2 and 3: Many thanks for this insight. We have performed substantial language revisions.  We hope that the reviewer agrees that the manuscript is improved as a result.

-Among adipokines, do the authors have data on leptin? This would be useful as it is a strong determinant of BP and also of glucose homeostasis.

Re 4. Unfortunately we do not have data on leptin. This absence has been added to the study limitations subsection.

-The recruitment period is not stated.

Re 5. We apologise for this oversight.  This information has now been added to the subsection “2.1. Study group”.

-It is not clear why and how the variables included in the network analysis were chosen.

Re 6. This information has now been added to the end of the Introduction section. We hope that the Reviewer would find it to be clear.

-The discussion should begin with a brief summary of the main findings, not with the results of another paper.

Re 7. We agree with the reviewer and have now amended the discussion as they have suggested. .

Unfortunately, although of potential clinical interest, the key message of the paper is not clear and the authors focus too much on the possible significance of each risk factors without offering a comprehensive interpretation of the results.

Re 8. We have amended the manuscript and tried to explain how the findings  are clinically relevant in the second subparagraph in the discussion section. We hope that is more clear now.

Reviewer 3 Report

In this interesting paper,

Kujawska et al. carried out a cross-sectional study with a total sample of 31 patients diagnosed with poorly managed hypertension and 18 normotensive.

They analyzed the network relationship between cognitive function, depressive symptom intensity, body composition, proxies of cognitive reserve, trophic factor, adipokines and myokines, physical performance and blood pressure.The results showed that Neighbourhood connectivity and stress were significantly higher in network of hypertensive group comparing to normotensive group.The authors found this to be of interest and conclude from their results that In older subjects with poorly managed hypertension, dyshomeostasis was observed, comparing to normotensive subjects.

The experimental approaches are sound and state of the art. In their results and discussion sections the authors delineate the importance of monitoring hypertension in older patients, which could beneficial for this patients. However, there are considerable flaws in the experimental design and a number of major issues that should be addressed, as outlined in detail below.

Major comment:

  1. The two compared groups are not balanced in the normal group there are 18 normotensives and in the poorly managed hypertension 31 patients. The authors should discuss this limitation in detail and maybe could increase the number of normotensive patients.

  1. The other big issue is that in both groups only 1 and 2 males are included. This is also a major limitation of this study and should be discussed.

  1. Additional blood parameters like LDL, HDL, triglycerides etc. should which have an influence on blood pressure should also added to table A1

Author Response

In this interesting paper,

Kujawska et al. carried out a cross-sectional study with a total sample of 31 patients diagnosed with poorly managed hypertension and 18 normotensive.

They analyzed the network relationship between cognitive function, depressive symptom intensity, body composition, proxies of cognitive reserve, trophic factor, adipokines and myokines, physical performance and blood pressure.The results showed that Neighbourhood connectivity and stress were significantly higher in network of hypertensive group comparing to normotensive group.The authors found this to be of interest and conclude from their results that In older subjects with poorly managed hypertension, dyshomeostasis was observed, comparing to normotensive subjects.

The experimental approaches are sound and state of the art. In their results and discussion sections the authors delineate the importance of monitoring hypertension in older patients, which could beneficial for this patients. However, there are considerable flaws in the experimental design and a number of major issues that should be addressed, as outlined in detail below.

Major comment:

  1. The two compared groups are not balanced in the normal group there are 18 normotensives and in the poorly managed hypertension 31 patients. The authors should discuss this limitation in detail and maybe could increase the number of normotensive patients.

 Re 1: We are grateful for the Reviewer’s comment. This information has now been added to the “4.5. Study limitations” subsection.

  1. The other big issue is that in both groups only 1 and 2 males are included. This is also a major limitation of this study and should be discussed.

  Re 2: Many thanks for this insight. This limitation has been also added to the study limitations subsection now.

  1. Additional blood parameters like LDL, HDL, triglycerides etc. should which have an influence on blood pressure should also added to table A1

  Re 3: As requested, information on need of additional parameters measured has also been added to study limitations subsection.

Round 2

Reviewer 2 Report

The Authors have improved the quality of the manuscript by clarifying many of the aspects of the network analysis.

Reviewer 3 Report

The authors have adequately addressed most, but not all, of the comments raised and in my opinion have significantly improved the manuscript. I think that the manuscript now merits publication in IJERPH.

This manuscript is a resubmission of an earlier submission. The following is a list of the peer review reports and author responses from that submission.

Round 1

Reviewer 1 Report

In general, t his manuscript is not clear, and not presented in a well-structured manner. Approximately 40% of references have more than 5 years. Is not scientifically sound and is the experimental design is not appropriate to test the hypothesis. There is no sample size calculation to support the hypothesis. There is a big lack details in the methods section.

ABSTRACT: Methodology with missing data; poorly described results; conclusion, objectives, and title without connection.

INTRODUCTION: There is no connection between the sentences; lack of clarity on what you want to say; the disease that the title focuses on is hypertension. However, this disease is not well contextualized in the introduction; “Recent studies”, however, he only cites one; the population is elderly; this is not clear either in the title or in the abstract.

MATERIALS AND METHODS: Body composition: Reported height, not measured, which leads to bias. There is no information on which equation was used in the body composition calculations; lack of clarity regarding the specification of bioimpedance; abbreviations without meaning; lack of reference in Blood pressure examination methods; Who evaluated the Emotional state assessment? Biochemical examination evaluation without parameters

RESULTS:

The final sample is described both in the methodology and in the results.

Authors describe information similar to those in table 1.

Title of table 1 refers to patients with a history of hypertension, however the table shows characteristics of hypertensive and normotensive individuals.

There are several variables in table 1, which were not described in the methodology.

Lack of standardization in the table.

The methodology says that “To examine between-group differences Mann–Whitney U or independent T-tests were used depending on assumptions met”, however, there is no indication of which variable used each test.

Confusing and poorly described results.

Reviewer 2 Report

The text has a broad but confusing objective. It remains to determine the research questions of the study. The study was carried out with a very small sample of people, which seriously compromises the inferences and repercussions of the work in clinical practice. The sample between the two groups was not matched, at least by sociodemographic data, which may also have influenced the results. Furthermore, the assessment of physical activity and blood pressure needs to be reviewed. Blood pressure measurement must follow international standards and, just as physical activity was evaluated only for days, without considering time and intensity. The results in networking format were not clear and the text does not bring the implications for clinical practice.